# Academic Passion and Subjective Well-Being among Female Research Reserve Talents: The Roles of Psychological Resilience and Academic Climate

**DOI:** 10.3390/ijerph20054337

**Published:** 2023-02-28

**Authors:** Zijun Yin, Bin Xuan, Xiaoyan Zheng

**Affiliations:** School of Educational Science, Anhui Normal University, Wuhu 241000, China

**Keywords:** academic passion, psychological resilience, academic climate, subjective well-being, female research reserve talents

## Abstract

This study aimed to investigate the relationship between academic passion (AP) and subjective well-being (SWB), along with the mediating role of psychological resilience (PR) and the moderating role of academic climate (AC), among Chinese female research talent in reserve. A convenience sampling method was used to select 304 female master’s degree students from several universities in the central region of China a questionnaire survey. The results show that: (1) AP has a positive predictive effect on the SWB of female reserve research talents; (2) PR partially mediated the relationship between the AP and SWB of female reserve research talents; (3) AC moderated the relationship between the AP and SWB of female reserve research talents. Thus, the findings of this study support a moderated mediation model that explores the relationship between AP and SWB for female research backups, with PR as the mediating variable and AC as the moderating variable. These findings provide a new perspective with which to explore the mechanisms influencing the subjective well-being of female research reserves.

## 1. Introduction

Based on Nature’s 2019 global public data on Ph.D. students, some scholars have found that Chinese female Ph.D. students are less satisfied with their doctoral studies, have a lower sense of academic career identity, and are less likely to choose an academic career compared to their male counterparts [1,2]. Similarly, due to the influence of traditional gender concepts, female master’s students not only have to experience the pressure of academic research, but also have to bear the expectations of society, family, and individuals; and suffer psychological pressure from financial, academic, employment, and marital considerations [3]. Previous studies have focused on the female doctoral student population, but the master’s student population is also a female research reserve, and they are new to research and academia, full of curiosity, and approaching research and academia with enthusiasm [1,2,3]. As the new generation of female research reserve talents, female master’s students are in the nascent stage of scientific research. Based on the social reality in China, women in postgraduate studies in China are generally in the age range of 20 to 30 years old. Hence, for the purposes of this study, we focused on female research reserve talents who were between the ages of 20 and 30 and were undergoing systematic, professional academic training as Chinese female master’s students.

The issue of women’s subjective well-being has received much attention from scholars both nationally and internationally [4]. Some researchers have pointed out that whether women can experience happiness in their research work is an important reason for their future persistence, or lack thereof, in research positions [4]. Therefore, this study focuses on the causes and mechanisms that affect the subjective well-being of female research reserve talents to motivate expect more women to devote themselves to research and call on the government, society, and families to provide them with more protection, support, and assistance.

### 1.1. The Role of Academic Passion in Subjective Well-Being

Passion is mostly applied in work domain research, and general passion theory considers passion as a strong and positive affective state that inspires and promotes individual thoughts and behaviors [5]. Based on this theory, we extend the definition of passion to academic passion, which can be defined as a state of strong emotion in an individual towards academic research, accompanied by cognitive and behavioral manifestations of high self-efficacy. Moreover, some researchers have proposed the theory of motivation, in which an individual’s passion for something is a powerful inner energy that drives the individual to take steps to do this thing well [5,6,7]. In short, it can be said that passion is a positive intrinsic force of motivation. Thus, we speculate that academic passion, as a strong intrinsic motivation, can lead individuals to spend a great deal of time and energy to better accomplish their academic research work. Some studies have found that positive passion can effectively improve an individual’s level of psychological well-being [8], which often includes subjective well-being. In addition, it has been shown that students’ passion for studies can enhance their happiness in life [9]. Based on the above discussion, we focus on the impact that academic passion has on the subjective well-being of Chinese female research reserve talents and propose the following hypothesis.

**Hypothesis 1** **(H1).**Academic passion positively affects the subjective well-being of Chinese female research reserve talents.

### 1.2. The Mediating Rule of Psychological Resilience

Psychological resilience (PR) is the ability of individuals to adapt to the complex circumstances around them; to cope positively with various situations in life, including non-traumatic and traumatic negative experiences; and to maintain a positive mindset [10,11]. Psychological resilience is also considered to be a dynamic procedure which is mainly manifested in facing difficulties [12]. Positive affectation is a core mechanism of the psychological resilience process [13]. Three major elements, personal, family, and environmental factors, have been categorized by researchers as factors influencing psychological resilience, and personal factors are especially crucial [10]. Studies have confirmed that factors such as an individual’s open and extroverted personality, self-efficacy, gender, and positive effect and hopes can support psychological resilience [14]. Fredrickson’s broaden-and-build theory states that a positive effect can broaden the mind and help individuals establish stability and sustainable psychological resources, such as psychological resilience, contributing to their success and happiness [15]. The dualistic model of passion (DMP) defines passion as a strong disposition in individuals toward enjoying an activity, considering it important and meaningful, and investing a lot of time and energy into it; and the activity is part of the identity in such cases [16]. Based on the broaden-and-build theory and DMP [15,16], we believe that the stronger the individual’s academic passion and the stronger the attitude towards academic research, the more positive emotions can be maintained, not only to cope with difficulties but also to dissipate negative emotions, such as frustration. Hence, we believe that academic passion can enhance an individual’s ability to be psychologically resilient. Furthermore, psychological resilience supported by academic passion makes a vital contribution to an individual’s subjective well-being and psychological well-being. There are numerous empirical studies that have confirmed the positive impact of psychological resilience on multiple aspects of an individual’s mental health and well-being [17,18,19,20]. Furthermore, psychological resilience has been shown to directly and positively predict individuals’ subjective well-being [21,22]. Following the above discussion, it is reasonable to speculate that psychological resilience can play a mediating role in the relationship between academic passion and subjective well-being.

**Hypothesis 2** **(H2).**Psychological resilience mediates the relationship between Chinese female research reserve talents’ academic passion and subjective well-being.

### 1.3. The Moderating Rule of Academic Climate

The term “climate” was first proposed by Lewin Climate as an employee’s evaluation and perception of the environment, as an important predictor of employee motivation and behavior, and it can directly or indirectly affect employee performance [23]. The academic climate proposed in this study refers to the inner qualities of universities and the cultural heritage condensed through various academic activities, which can implicitly influence the motivation, academic ethics, and academic passion of members within a university, and also have different degrees of influence on individual well-being [24]. Permarupan et al. believe that a positive organizational climate helps stimulate individuals’ work passion [25]. The academic climate is highly correlated with academic passion, and a positive academic climate helps to stimulate individuals’ academic passion and make them more active in academic research, which as we already have learned, can directly affect the academic performance of university students [26]. As a result, in a good academic atmosphere, individuals can gain a better sense of self-efficacy in efficient and high-quality academic research [23], thereby enhancing their subjective well-being. In addition, a study of freshman undergraduates at an Italian university has shown that academic climate can significantly affect individuals’ well-being [27]. Thus, we speculate that the academic climate can effectively influence individuals’ subjective well-being. Furthermore, social exchange theory believes that when individuals sense concern and support in the environment, it can stimulate their motivation and inspire them so that they will be more proactive in devoting themselves to the environment to maximize the benefits for themselves and the environment and achieve a win–win situation for both individuals and the environment [24]. This means that individuals can obtain material or spiritual benefits, such as subjective well-being. On the other hand, we mentioned above that passion can significantly influence subjective well-being, and from an ecological perspective, the relationship between the two can likewise receive mediation or moderation by a variety of factors. The academic climate, as an environmental variable, may have an impact on the relationship between the two variables. Hence, academic climate may function as a moderating factor. It has been established that a task-performance academic climate moderates the relationship between achievement motivation and academic achievement [28]. It is argued that academic climate can act as a moderating variable involved in the moderation between academic passion and subjective well-being.

**Hypothesis 3** **(H3).**Academic climate moderates the relationship between Chinese female research reserve talents’ academic passion and subjective well-being.

### 1.4. The Present Study

Based on the above hypotheses, we set up a moderated mediation model to explore three questions. The first issue is the role of academic passion in the subjective well-being of Chinese female research reserves. The second issue is the mechanism by which psychological resilience affects the relationship between academic passion and subjective well-being. The third issue is the mechanism by which the academic climate influences the relationship between academic passion and subjective well-being. The proposed model is presented in Figure 1.

## 2. Materials and Methods

### 2.1. Participants

A convenience sampling method was employed to obtain participants on 2 January 2023 from several universities in central China. A total of 334 female postgraduates were selected as participants. The participants who filled in the questionnaire randomly and whose variable scores were more than plus or minus 3 standard deviations were deleted. Finally, 304 valid responses were obtained—an effective recovery rate of 91.02%. The average age of participants is 24.06 years old (ranging from 21 to 30, SD = 1.67). About 56% of them were in the first grade, 27% of them were in the second grade, and 17% of them were in the third grade. About 37% of them wanted to engage in academic research in the future, and about 63% of them did not want to. About 41% of them had participated in scientific research projects, and 59% had not. Furthermore, 26% of them had published academic papers, and 74% had not. All subjects signed an electronic informed consent form before completing the electronic questionnaire.

### 2.2. Measures

#### 2.2.1. Academic Passion Scale

The academic passion scale was adapted from Vallerand’s passion rating scale [29], revised by Wang [30], and was used to measure the degree of the academic passion of individuals. The scale consisted of 10 items, and we adopted Likert 6-point scoring (1 to 6 indicating completely inconsistent to completely consistent, respectively). This scale has been shown to have good consistency in Chinese college students [30]. In our study, Cronbach’s α coefficient for the academic passion scale was 0.916.

#### 2.2.2. Psychological Resilience Scale

The psychological resilience scale was adapted from Connor–Davidson Resilience Scale (CD-RISC). The Chinese-translated version of this scale measures the degree of psychological resilience of individuals and was proven to have good consistency in Chinese subjects [31,32]. The scale consists of 10 items, and it uses Likert 5-point scoring (1 to 5, respectively, never to all the time). In our study, Cronbach’s α coefficient for the psychological resilience scale was 0.925.

#### 2.2.3. Academic Climate Scale

The Chinese-translated version of the academic climate scale was developed by Wang [24]. It measures the individuals’ subjective perceptions of the academic climate at their universities, and it was proven to have good consistency in Chinese subjects [24]. The scale consists of 6 items, and it uses Likert 5-point scoring (1 to 5, strongly disagree to strongly agree, respectively). In our study, Cronbach’s α coefficient for the academic climate scale was 0.927.

#### 2.2.4. Subjective Happiness Scale

The Subjective Happiness Scale (SHS) was developed by Lyubomirsky et al. [33]. The Chinese-translated version of the SHS was translated by our research group and was used to measure the degree of the subjective happiness of each individual. The scale consisted of 4 items, and it uses Likert 7-point scoring (1 to 7, strongly disagree to strongly agree, respectively). This scale had been shown to have good consistency in Chinese college students [34]. In our study, Cronbach’s α coefficient for the Subjective Happiness Scale was 0.756.

### 2.3. Procedure

First, we constructed a moderated mediation model based on the theoretical model and the results of previous studies. Second, we invited female master’s students from several cities in central China to fill out an electronic questionnaire. We obtained their informed consent electronically before the questionnaire was written and ensured that they were aware that the questionnaire was anonymous and voluntary, that they could withdraw freely at any time, and that the data results were confidential. Third, the female postgraduates were asked to answer independently according to their actual situation. It took approximately 5 min to complete the questionnaire. Then, we input the collected data and filtered out invalid data.

### 2.4. Data Analysis

IBM SPSS 24.0 was used to perform a common method deviation test, a descriptive statistical analysis, a correlation analysis, and a regression analysis. Model 4 in the SPSS macro program PROCESS plug-in was used to perform a mediating effect test, and model 5 was used to perform a moderated mediated effect test [35]. In this case, the bootstrap method was used to draw samples 5000 times and estimate 95% confidence intervals.

## 3. Results

### 3.1. Common Method Biases Tests

Harman’s single-factor test extracted five factors with eigenvalues greater than one. The first factor explained 36.393% of the total variance, which is below the recommended threshold of 40%. This suggests the common method bias was unlikely to confuse the data analysis results.

### 3.2. Participants’ Characteristics

A total of 304 female postgraduates participated and effectively filled out the questionnaire. The basic characteristics of the participants are shown in Table 1.

### 3.3. Descriptive Analysis and Correlations between Overall Variables

Table 2 shows the mean (*M*), standard deviation (*SD*), and Pearson correlation of each variable. The total score for AP was 34.760 ± 8.815; for PR, it was 35.310 ± 6.093; for AC, was 22.640 ± 4.273; and for, it SWB was 19.430 ± 3.822.

The Pearson correlation analysis demonstrated that AP was positively correlated with PR (*r* = 0.496, *p* < 0.01), AC (*r* = 0.308, *p* < 0.01), and SWB (*r* = 0.425, *p* < 0.01). PR was positively correlated with AC (*r* = 0.321, *p* < 0.01) and SWB (*r* = 0.564, *p* < 0.01). AC was positively correlated with SWB (*r* = 0.401, *p* < 0.01).

### 3.4. Testing the Mediating Effect of Psychological Resilience

Firstly, multiple linear regression analysis showed that gender, age, grade, and research project experience had no significant influence on SWB, and future academic work intention had a significant influence on SWB. As a result, future academic work intention was included as a covariate in the regression analysis, mediating effects analysis, and moderation effects analysis.

Secondly, the hierarchical regression model and PROCESS’s macro model 4 were used to analyze the mediating role of PR in the relationship between AP and SWB. After controlling for future academic work intention, AP can significantly positively predict PR (*β*_1_ = 0.507, *t* = 8.838, *p* < 0.001) and SWB (*β*_2_ = 0.432, *t* = 7.220, *p* < 0.001), and PR can significantly positively predict SWB (*β*_3_ = 0.548, *t* = 11.258, *p* < 0.001). When AP and PR were used in the regression equation together, the predictive effect of AP on SWB was still significant (*β*_4_ = 0.194, *t* = 3.233, *p* < 0.01), and PR also still had a significant positive predictive effect on SWB (*β*_5_ = 0.469, *t* = 8.721, *p* < 0.001). This was manifested in PR partially mediating the relationship between AP and SWB.

Thirdly, PROCESS’s macro model 4 was used to analyze the mediating role of PR in the relationship between AP and SWB. After controlling for future academic work intention, the Bootstrap method test with percentile bias correction indicated that AP had a direct effect on SWB, with *β*_6_ = 0.084, Boot SE = 0.026, and 95% CI = (0.033, 0.135). PR had a significant mediating effect between AP and SWB, with *β*_7_ = 0.103, Boot SE = 0.021, and 95% CI = (0.064, 0.146). The contribution rates of indirect effects in the total effect were β_7_/(*β*_6_ + *β*_7_) = (0.103/0.187) = 55.080%.

Therefore, Hypothesis 2 was supported. Table 3 demonstrates the direct effect of academic passion on subjective well-being, the indirect effect of psychological resilience on subjective well-being, and the total effect of academic passion and psychological resilience on subjective well-being.

### 3.5. The Moderating Effect Analysis

PROCESS’s macro model 5 was used to analyze the moderating role of academic climate. As shown in Table 4, the interaction terms between AP and AC (*β*_8_ = 0.012, *p* < 0.01) had a significant predictive influence on the SWB of female postgraduates after controlling for future academic work intention. This suggests that the academic climate moderated the association between AP and SWB. Therefore, Hypothesis 3 was supported.

A simple slope analysis was used to further the visual analysis of the moderating role of the academic climate. The results show that the predictive effect of AP on a good academic climate’s SWB was significantly higher than that of AP on a poor academic climate’s SWB (high academic climate: *β*_9_ = 0.103, *p* < 0.001; low academic climate: *β*_10_ = 0.003, *p* > 0.05). Figure 2a shows the moderating role of the academic climate between AP and SWB. Figure 2b illustrates a mediated model with moderation.

## 4. Discussion

The purposes of this study were to examine the relationships among academic passion, psychological resilience, academic climate, and subjective well-being among Chinese female master’s degree students; and to explore the mediating role of psychological resilience and the moderating role of the academic climate. First, the results show that there is a significant correlation between all these variables. Second, academic passion was proven to have a significant effect on subjective well-being, and psychological resilience played a partial mediating role in their relationship. Finally, our study found that academic climate moderated the relationship between academic passion and subjective well-being. Specifically, academic passion had a greater effect on the subjective well-being of female master’s students in a high academic climate compared to a low academic climate. These results have important implications and practical value for further research and exploration of the subjective well-being of female research postgraduates.

### 4.1. Discussion of the Correlations between the Variables

The results of the relevant studies confirmed that academic passion, psychological resilience, and academic climate were positively correlated with subjective well-being. We have not found any other scholars exploring the relationship between academic passion and subjective well-being, but one study found a significant effect of teachers’ positive work passion on their subjective well-being [36]. Another study found a significant effect of undergraduates’ academic passion on their happiness [37], which remains largely consistent with our findings. Other studies found that an individual’s level of psychological resilience can affect his or her subjective well-being [22], which is consistent with our findings. Apart from this, no studies were found to examine the effect of an academic climate on subjective well-being, but some studies have verified that a poor campus climate can significantly reduce the level of subjective well-being of adolescents [38], which indirectly validates our findings. This also illustrates the contribution of our study regarding the factors influencing the subjective well-being of female science and technology reserves.

### 4.2. The Mediating Role of Psychological Resilience

The mediating effect results found that academic passion affects the subjective well-being of female research reserves not only directly but also indirectly, through the effect of psychological resilience. It had been suggested, based on evolutionary theory [39] and organismic theories of humans’ inherent desire for growth [40], that when an individual’s basic psychological needs are met, it leads to a sense of subjective well-being [41]. Importantly, the fulfillment of basic and psychological needs was additive for SWB, indicating that each need promotes subjective well-being independently of other needs [41]. When an individual’s basic psychological needs are satisfied, he or she pursues the need for self-actualization. To some extent, we believe that academic passion is a need for self-fulfillment, and therefore we have demonstrated both theoretically and empirically the positive impact of academic passion on subjective well-being. Individuals who wanted to achieve their growth needs (in this article, their academic and scientific passions specifically) encountered great challenges and difficulties, and the courage and determination to overcome them became invaluable, known in psychology as psychological resilience [10,11]. The mental state theory of subjective well-being elaborates that an individual’s cognitive, attentional, and mental state processes affect the subjective perception of well-being [41], and that psychological resilience is a positive mental state [10,11], the level of which affects an individual’s subjective well-being to varying degrees. Female master’s students are new to research and approach it with a sense of the unknown and curiosity. When they are truly engaged in scientific research (i.e., the stronger the individual’s academic passion), they will also be more resilient to face the obstacles they encounter in research (i.e., the higher the individual’s psychological resilience) and will naturally feel well-being that research brings them (i.e., the higher the individual’s psychological resilience). Based on the above discussion, we have explained the mediating role of psychological resilience in the relationship between academic passion and subjective well-being based on the perspectives of evolutionary theory, organismic theories of humans’ inherent desire for growth, and the mental state theory of subjective well-being.

### 4.3. The Moderating Role of Academic Climate

The moderating effect results found that a positive academic climate can moderate the relationship between academic passion and subjective well-being. Specifically, the more positive the academic climate perceived by female research candidates, the higher their level of subjective well-being. The biological theory of subjective well-being explains why most daily events affect one’s SWB for a short time, and that there is an eventual return to baseline [42]. This phenomenon is known as set-point theory, in which individuals have their own biologically determined set points to which they return over time. Later, scholars expanded on set-point theory, arguing that everyone has their subjective well-being tendencies, but that differences in environmental climate and different perceptions of environmental climate could affect subjective well-being, so there was no fixed set point for everyone. Based on the set-point theory and its extensions, we argue that academic passion has a direct impact on the subjective well-being of women in science and technology reserves, and each individual’s academic passion ‘set-point’ was different and was influenced by the academic research environment in which she operated.

### 4.4. Limitations and Future Research Directions

Our study had some limitations, as follows. First, our conclusion was based on cross-sectional data, which cannot investigate causal relationships. Second, we recruited subjects who were female master’s students, and our focus was on female research reserve talents, so the group of subjects surveyed was quite homogeneous. Finally, the direct and indirect mechanisms for the relationship between academic passion and subjective well-being have been explored in our study, and other indirect explanatory pathways may exist. 

Future studies could be conducted as follows. First, future research could be based on longitudinal studies, behavioral experiments, and cognitive neuroscience instruments to explore the causal relationships between the above variables. Second, future research could enrich the survey groups of female research reserves, for example, by investigating the subjective well-being of groups of female Ph.D. students and young female university teachers. Third, future research could explore the effects of different types of academic passions on the subjective well-being of female research talent in reserve, e.g., harmonious and compulsive academic passions, and could explore the indirect mechanisms of the relationship from the perspectives of gender stereotypes and socioeconomic status. Fourth, future research could continue to explore in depth the relationship between the four variables of academic passion, psychological resilience, academic climate, and subjective well-being in the male master’s student cohort, and compare the results of the model for the male and female master’s student cohorts.

## 5. Conclusions

Despite the above limitations, to the best of our knowledge, our study was the first to explore the relationship between academic passion and the subjective well-being of female science and technology candidates. Our study found a significant correlation among academic passion, psychological resilience, academic climate, and subjective well-being. In addition to this, psychological resilience mediated the relationship between academic passion and subjective well-being. The academic climate played a moderating role in the relationship between the two. Specifically, the positive relationship between academic passion and subjective well-being was stronger for female research candidates in a good academic climate than for female research candidates in a poor academic climate.

Our findings provide a new perspective on the mechanisms influencing the subjective well-being of the female research talent group and provide theoretical value in calling for more women to engage in scientific research. The government, society, schools, and mentors should give them a fairer, more harmonious, and a supportive academic atmosphere; female research prospects themselves should fully mobilize their passion for academic research, sharpen their patience and perseverance in the face of difficulties, and experience the happiness and joy of the academic frontier in the ocean of science. In short, scientific research needs women, and women can achieve outstanding results in academia. Hence, we hope to see more female talent emerge. Last but not least, we would like to say that scientific research needs women’s strength; and only passion can withstand the long years, and only by persevering with the passion of academia in one’s heart and keeping one’s the original intentions can one pass through the long road of scientific research.

## Figures and Tables

**Figure 1 ijerph-20-04337-f001:**
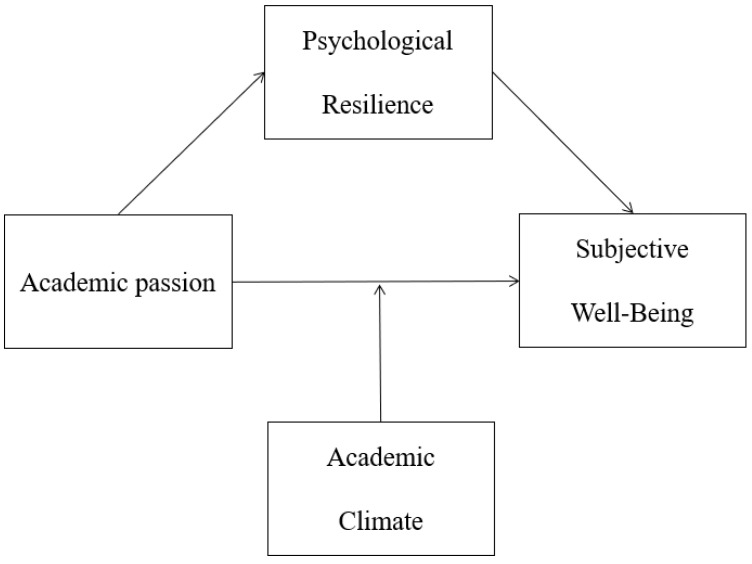
The proposed model.

**Figure 2 ijerph-20-04337-f002:**
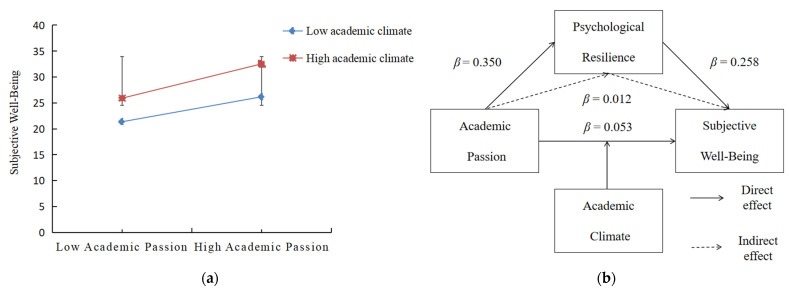
**A** diagram of the slope of the moderation effect and the mediated model with moderation. (**a**) The moderating role of the academic climate between AP and SWB. (**b**) A mediated model with moderation.

**Table 1 ijerph-20-04337-t001:** Demographic characteristics of female postgraduates.

Characteristic	Category	*N* = 304	%
Gender, *n* (%)	Female	304	100
Age, *M* (*SD*)	Age	24.060 (1.659)	
Grade	Grade 1	170	55.900
	Grade 2	83	27.300
	Grade 3	51	16.800
Future academic work intention	Want	111	36.500
	Do not want	193	63.500
Research project experience	Participate	125	41.100
	Did not participate	179	58.900
Published academic paper	Published	80	26.300
	Unpublished	224	73.700

**Table 2 ijerph-20-04337-t002:** Mean, standard deviation, and Pearson correlation of each variable.

Variables	*Mean* ± *SD*	AP	PR	AC	SWB
AP	34.760 ± 8.815	1			
PR	35.310 ± 6.093	0.496 **	1		
AC	22.640 ± 4.273	0.308 **	0.321 **	1	
SWB	19.430 ± 3.822	0.425 **	0.564 **	0.401 **	1

Abbreviations: AP, academic passion; PR, psychological resilience; AC, academic climate (AC); SWB, subjective well-being. ** *p* < 0.01.

**Table 3 ijerph-20-04337-t003:** Mediating effects and 95% confidence intervals estimated by the bootstrap method.

Effect	Estimate	95% CI			Percentageof Effect
		Lower	Upper	*p*	
Direct effect	0.084	0.033	0.135	**	45.920%
Indirect effect	0.103	0.064	0.146	***	55.080%
Total effect	0.187	0.136	0.238	***	100.000%

Notes: ** *p* < 0.01; *** *p* < 0.001.

**Table 4 ijerph-20-04337-t004:** The moderating role of the academic climate in the relationship between AP and SWB.

Predictive Variable	Model 1 (Criterion: SWB)		Model 2 (Criterion: PR)		Model 2 (Criterion: SWB)		Model 3 (Criterion: SWB)	
	*β*	*t*	*β*	*t*	*β*	*t*	*β*	*t*
FAWI	0.013	0.215	0.274	0.378	0.022	0.051	−0.037	−0.091
AP	0.432	7.220 ***	0.350	8.838 ***	0.084	3.233 **	0.053	2.078 *
PR					0.294	8.721 ***	0.258	7.818 ***
AC							0.221	5.067 ***
AP × AC							0.012	2.970 **
R2	0.181		0.247		0.347		0.406	
F	33.264 ***		49.236 ***		53.057 ***		40.810 ***	

Abbreviations: FAWI, future academic work intention; AP, academic passion; PR, psychological resilience; AC, academic climate (AC); SWB, subjective well-being. * *p* < 0.05; ** *p* < 0.01; *** *p* < 0.001.

## Data Availability

Data are available from the corresponding authors upon reasonable request.

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
