# Peer review of "Academic Passion and Subjective Well-Being among Female Research Reserve Talents: The Roles of Psychological Resilience and Academic Climate"

_ijerph, 2023, doi:10.3390/ijerph20054337_

Round 1

Reviewer 1 Report

Academic Passion and Subjective Well-Being among Female Research Reserve Talents: The Role of Psychological Resilience and Academic Climate

Comments:

1.       The manuscript focuses on the impact of academic passion on the subjective well-being of Chinese female research reserve talents. The succinct discussion of the paper is commendable. In pithy words, the juxtaposing of concepts –  AP, SWB,  PR and AC – was commendably performed in a well-woven narrative.

2.       Findings are empirically sound and prudently tested. Interpretation of the data coheres with the statistical findings. The current findings are even combined with previous findings to strengthen the correlation of the variables.

3.       The conclusion answers the research questions stated and the hypotheses, soundly tested and measured, carefully supported the statistical analysis of the data.

4.       A working definition of research reserve talent is in order to have a better grasp of whom we are talking about.  To simply say that “we focus in this study on Chinese female master's students aged between 20 and 30 years old who are currently undergoing systematic and specialized academic training” is vague.  I also suggest that you give a justification for choosing this group of people, and the age range as well, to test your hypotheses. Is there something unique about this group of people and their particular age range?

5.       Line 146: is this line correct? 27% of them were in the first (or should be second) grade.

6.       The Discussion is very robust as it is supported by the Results presented in the study.

7.       Very few literature studies were invoked in the study. Some studies are even dated as early as 1975. Consider updating the literature review as some studies may have surfaced between January (when the study was conducted) and now. This one may be insightful:  Spiridon, K. Investigation of the relationships between academic hardiness and passion for studies with undergraduates’ affect and happiness. SN Soc Sci 2, 201 (2022). https://doi.org/10.1007/s43545-022-00518-1

8.       Minor grammar lapses were observed.

Author Response

Response to Reviewer 1

Dear Reviewer,

We are grateful for your agreement to the revised manuscript (Manuscript ID ijerph-2210811). Your acceptance of our revised manuscript is very motivating and inspiring for us. Following your, editors’, and all reviewers’ comments, this paper has been considerably strengthened. Once again, please accept our thanks for providing us with the opportunity to resubmit our manuscript to the International Journal of Environmental Research and Public Health.

Thanks for all the help.

Prof. Bin Xuan

Reviewer 1 Comments

1. The manuscript focuses on the impact of academic passion on the subjective well-being of Chinese female research reserve talents. The succinct discussion of the paper is commendable. In pithy words, the juxtaposing of concepts – AP, SWB, PR and AC – was commendably performed in a well-woven narrative.

Answer: We are very grateful for your valuable and detailed comments, which have given us great encouragement and we will continue to improve the exposition of variable relationships in our thesis in the light of your comments.

2. Findings are empirically sound and prudently tested. Interpretation of the data coheres with the statistical findings. The current findings are even combined with previous findings to strengthen the correlation of the variables.

Answer: We are very grateful for your valuable and detailed comments, which have given us great encouragement, and we will continue to improve the discussion of the data analysis and results in our paper based on your comments.

3.The conclusion answers the research questions stated and the hypotheses, soundly tested and measured, carefully supported the statistical analysis of the data.

Answer: We are very grateful for your valuable and detailed comments, which have given us great encouragement, and we will continue to improve the discussion of the data analysis and results in our paper based on your comments.

4. A working definition of research reserve talent is in order to have a better grasp of whom we are talking about. To simply say that “we focus in this study on Chinese female master's students aged between 20 and 30 years old who are currently undergoing systematic and specialized academic training” is vague. I also suggest that you give a justification for choosing this group of people, and the age range as well, to test your hypotheses. Is there something unique about this group of people and their particular age range?

Answer: We are very grateful for your valuable and detailed comments. Your suggestions have helped us to think more carefully and elaborate on why our study focused on a group of female Masters’s students.

We have accepted the reviewer's recommendation. And we have added " Based on Nature's 2019 global public data on Ph.D. students, some scholars have found that Chinese female Ph.D. students are less satisfied with their doctoral studies, have a lower sense of academic career identity, and are less likely to choose an academic career compared to their male counterparts [1-2]. However, due to the influence of traditional gender concepts, some researchers suggest that compared to male master's students, they not only have to experience the pressure of academic research, but also have to bear the expectations of society, family and individuals, and suffer psychological pressure from financial, academic, employment and marital aspects [3]. Previous studies have focused on the female doctoral student population, but have neglected the master's student population, which is also a female research reserve and is at a time when they are new to research and academia, full of unknowns and curiosity, and approaching research and academia with enthusiasm, which also deserves our attention [1-3]. As the new generation of female research reserve talents, female master's students are in the nascent stage of scientific research. Based on the social reality in China, the age of women in postgraduate studies in China is generally distributed in the age range of 20 to 30 years. Hence, for the purpose of this study, we focus on female research reserve talents who are between the ages of 20 and 30 and are currently undergoing systematic, professional academic training as Chinese female master's students." to explain the question "why was our study focused on a group of female masters students." (Line 26~43, Page 1; Marked in red) in the introduction of our paper.

5. Line 146: is this line correct? 27% of them were in the first (or should be second) grade.

Answer: We are very grateful for your valuable and detailed comments. And we have rechecked the data and made changes in response to your comments, and added " About 56% of them were in the first grade, 27% of them were in the second grade, and 17% of them were in the third grade. " (Line 159~161, Page 4; Marked in red) in the introduction of our paper.

6. The Discussion is very robust as it is supported by the Results presented in the study.

Answer. We are very grateful for your valuable and detailed comments, which have given us great encouragement, and we will continue to improve the discussion section of our paper based on your comments.

7. Very few literature studies were invoked in the study. Some studies are even dated as early as 1975. Consider updating the literature review as some studies may have surfaced between January (when the study was conducted) and now. This one may be insightful:  Spiridon, K. Investigation of the relationships between academic hardiness and passion for studies with undergraduates’ affect and happiness. SN Soc Sci 2, 201 (2022). https://doi.org/1007/s43545-022-00518-1.

Answer: We are very grateful for your valuable and detailed comments. Your suggestions have helped us to check and update the details of this paper references more closely. What’s more, we have accepted the reviewers' suggestions and added these references below to the references section of our paper, as follows.

1. Wang, W.P.; Yuan J.; Yang J.; Liu H.Q. Is it true that women are less satisfied with their doctoral studies? ---An empirical analysis based on data from Nature's 2019 Global Survey of Doctoral Students. Higher Education Exploration 2021, 7, 47-56. (Line 415~416, Page 10; Marked in red)

2. Central People's Government of the People's Republic of China. Exploring the impact and mechanisms of motivation for doctoral study on non-academic career choices of Ph.D. students --- An empirical study based on the 2019 Nature Global Ph.D. Student Survey. Higher Education Exploration 2022, 4, 67-74. (Line 417~419, Page 10; Marked in red)

3. Liao, H.P.; Wang W.L. Countermeasures and Suggestions on Promoting the Healthy "Growth” of Female Postgraduate in Universities from the Perspective of Social Gender. Journal of Hunan University of Science & Technology (Social Science Edition) 2019, 22(05), 157-167. (Line 420~422, Page 10; Marked in red)

38. Spiridon, K. Investigation of the relationships between academic hardiness and passion for studies with undergraduates’ affect and happiness. SN Social Sciences 2022, 2, 201. (Line 489~490, Page 11; Marked in red)

8. Minor grammar lapses were observed.

Answer: We are very grateful for your valuable and detailed comments. Your suggestions have helped us to check the grammatical details of the thesis more carefully. And we have checked and revised the grammatical expressions throughout the paper and the revised words and grammatical issues have been marked in red in the revised manuscript.

Reviewer 2 Report

The article investigates  the relationship between academic passion (AP) and subjective well-being (SWB), as well as the mediating role of psychological resilience (PR) and the moderating role of academic climate (AC) among Chinese female research talent in reserve.

The research is quite innovative and adds to previous reserach done on motivation enhancement amog colleges students. The authors centered on female master's students in China. This is an innovative perspective.

Please look at my review. Try to use the present tense when you present the research rather than the past tense.

Also, please note my corrections with regard to the use of articles and the use of markers of contrast.

Well done!

Author Response

Response to Reviewer 2

Dear Reviewer,

We are grateful for your valuable feedback on the revised manuscript (Manuscript ID ijerph-2210811). We have carefully considered all statements in preparing our revision, and we think that the feedback resulted in valuable additions, changes, and improvements to our manuscript.

We are very impressed and grateful to see your careful word-by-word corrections, and it is an honor to have met such an expert reviewer as you. And we have provided a point-by-point response to the comments made by you. In each case, we begin by restating each of the comments. Immediately below each comment, we indicate how we addressed each issue and, if applicable, where the changes occur within the manuscript. Once again, please accept our thanks for providing us with the opportunity to resubmit our manuscript to the International Journal of Environmental Research and Public Health.

Thanks for all the help.

Prof. Bin Xuan

Reviewer 2 Comments

1. Errors in grammar and writing.

Answer: We are very grateful for your valuable and detailed comments and verbatim corrections. Your suggestions and corrections have helped us to check the grammatical details of the thesis more carefully. We have checked and revised the grammatical expressions throughout the thesis based on your suggestions, and the revised words and grammatical issues have been marked in red in the revised version. The details are as follows.

2. Definition of Terms and Problems with Thesis Presentation.

Answer: We are very grateful for your valuable and detailed comments and verbatim corrections. Your suggestions and corrections have helped us to check more closely the definitions of terms and problems with the presentation of the thesis. We have checked and revised the definitions of terms and the presentation of the thesis throughout the paper in accordance with your suggestions, and the changes are marked in red in the revised version. The details are as follows.

(1) Moreover, some researchers have proposed the theory of motivation, in which an individual's passion for something is a powerful inner energy that drives the individual to take steps to do this thing well [5-7]. (Line 58~60, Page 2; Marked in red)

(2) Psychological Resilience (PR) is the ability of individuals to adapt to the complex circumstances around them, to cope positively with various situations in life, including non-traumatic and traumatic negative experiences, and to maintain a positive mindset [10,11]. (Line 72~75, Page 2; Marked in red)

(3) About 56% of them were in the first grade, 27% of them were in the second grade, and 17% of them were in the third grade. (Line 159~161, Page 4; Marked in red)

(4) Table 3 demonstrates the direct effect of academic passion on subjective well-being, the indirect effect of psychological resilience on subjective well-being, and the total effect of academic passion and psychological resilience on subjective well-being. (Line 252~255, Page 6; Marked in red)
